Shelters can negatively affect growth and welfare in lumpfish if feed is delivered continuously

Johannesen Asa asajoh@fiskaaling.fo 1
Joensen Nakita E. 2
Magnussen Eyðfinn 2
1 Nesvik Marine Centre, Fiskaaling , Hvalvik , Faroe Islands
2 Department of Science and Technology, University of the Faroe Islands , Tórshavn , Faroe Islands
Toonen Robert
Electronic publication date: 2018 May 25
Publication date: 2018
Volume: 6
Electronic Location ID: e4837
Received 2018 Mar 12; Accepted 2018 May 4
Copyright: ©2018 Johannesen et al.
Copyright year: 2018
Copyright holder: Johannesen et al.
License: This is an open access article distributed under the terms of the Creative Commons Attribution License, which permits unrestricted use, distribution, reproduction and adaptation in any medium and for any purpose provided that it is properly attributed. For attribution, the original author(s), title, publication source (PeerJ) and either DOI or URL of the article must be cited.
License URL: https://creativecommons.org/licenses/by/4.0/

Keywords: Animal husbandry, Fish welfare, Aquaculture, Lumpfish, Cyclopterus lumpus

Funding: No external funding was received for this work. The two author-affiliated institutes covered the cost of all work and materials involved.

==============================
Due to the socioeconomic importance of salmon farming in the North Atlantic and the economic impact of sea lice in this industry, there is high demand for novel pest control methods. One such method is the use of cleaner fish to remove the lice from the salmon. A cleaner fish that has recently gained in popularity due to its ability to work in cold water, is the lumpfish (Cyclopterus lumpus). This fish varies in efficiency, but when mortality is low and cleaning optimal, the fish are successful in keeping parasite burdens low. However, there is some concern for the welfare of lumpfish in the industry, because mortality is often high. This is sometimes attributed to inadequate feeding and shelter. Here we compare growth, body condition, and fin health of fish reared for four weeks in a crossed treatment design crossing shelter availability (shelter vs none) and feed delivery method (manual meal time feeds and continuous automated feeding). In terms of weight gain, shelter availability interacted with feeding method, with fish that had access to shelters and were fed using automated feeders gaining less weight than other fish. Fin health was not affected, but body condition was lowered both by access to shelter and being fed continuously. The results indicate a need to carefully consider how feeding method and shelter use is combined, both in cages and during rearing on land.

Introduction

Aquaculture is a growing industry and already now outcompetes fisheries as a source of seafood protein (FAO, 2016). Because of increasing demand, some sectors in the industry are under pressure to produce ever increasing volumes of food. The Atlantic Salmon (Salmo salar) farming industry is one of the sectors, where production has increased rapidly in recent years (FAO, 2016). The increased production is at risk of impacting fish welfare (Turnbull et al., 2005; Jansen et al., 2012). One of the problems in the North Atlantic stems from two parasitic copepods, Lepeoptheiris salmonis (salmon louse) and Caligus elongatus (sea louse). These parasites are often referred to collectively as ‘sea lice’ and they have a large impact on the salmon farming industry, causing huge economic losses (Costello, 2009; Abolofia, Asche & Wilen, 2017).

Salmon lice consume mucous, skin, and blood from salmon (Costello, 2006), causing sub-lethal injury to salmon and in extreme cases, mortality (Skilbrei et al., 2013). To avoid this impact on both salmon production and welfare, farming companies and governments alike implement a range of rules and methods to keep infestation levels low. Historically, louse control methods have mostly consisted of chemical treatments and management practices designed to lower exposure to lice. While management practices are mostly sound, lice have proven to grow resistant to chemotherapeutants soon after they become available on the market (Aaen et al., 2015), so industry has been forced to find new ways of preventing accelerating louse infestations.

Many of the new louse prevention or removal methods are physical in nature. They involve either preventing louse access to salmon by using fine mesh “skirts” on cages or making the salmon swim at depths below where most of the free floating lice are (Frank et al., 2014; Stien et al., 2016), or they involve treatments using fresh water, warm water or pressurised water to flush the lice off the salmon (Grøntvedt et al., 2015; Lekang, Salas-Bringas & Bostock, 2016). All of these treatment and prevention methods are at risk of causing reduced salmon welfare or financial losses to farmers. Skirts can cause poor oxygenation (Stien et al., 2012) and “taking the surface away” from the salmon prevents natural behaviours (Barber, 2007). The various mechanical delousing methods cause stress and injury with resulting low appetite and mortality (reviewed in Bui et al., 2017). Some more inventive methods include a laser removal tool (Beck, 2015; Dumiak, 2017) and vegetable supplements in the feed (Jodaa Holm et al., 2016; O’Shea et al., 2017), but these are not yet commercially viable products. Finally, salmon farmers have been using cleaner fish, first a range of wrasse species (Fam. Labridae) starting in the 1980’s (Bjordal, 1988; Deady, Varian & Fives, 1995; Skiftesvik et al., 2014) and more recently, lumpfish (Cyclopterus lumpus) (Imsland et al., 2014a; Eliasen et al., 2018). Despite their generally hardy nature, lumpfish mortality on farms has been high and efficiency variable (Powell et al., 2017).

Lumpfish are a sedentary species with little buoyancy control due to the lack of swim bladder (Davenport, 1985). They use a sucker to attach to smooth surfaces and will generally only swim when necessary to avoid danger or to forage for food (Williams & Brown, 1991). Lumpfish that clean salmon will approach docile salmon, search for lice on the salmon, and pick the lice that they find off the salmon (Video S1). This behaviour is quite energy intensive, especially during times with strong currents or waves. In general, approximately 10% of lumpfish can be expected to have recently consumed lice at any one point in time on the Faroe Islands, though this is highly affected by seasonal availability of plankton causing a shift from 0 to 5% of lumpfish consuming lice when planktonic prey is available, to 15–20% when no planktonic prey is available (Eliasen et al., 2018). There is also some question as to the reliability of using lice found in the stomach as an indicator of how many lumpfish consume lumpfish because digestion times vary (Eysturskarð, Johannesen & Eliasen, 2016). The nutritional needs of lumpfish are not well documented, but given their prey choice, it is unlikely that lice meet those needs. However, it is possible that lumpfish prefer planktonic prey because they can employ a sit-and-wait strategy when planktonic prey is plentiful and simply stay attached to their shelters until the prey floats by in the current. There is no indication that lumpfish body condition is affected by louse burden in a salmon farm (Eliasen et al., unpublished data), so it is unlikely that lice play a major role in terms of nutrition for lumpfish.

Lumpfish are generally considered to use shelters, mainly for two purposes. They hide in or under them as an anti-predator behaviour and attach to them to rest (Imsland et al., 2014b; Imsland et al., 2015). This second use is related to their lack of swim bladder and the fact that their near spherical bodies are not well suited for rapid continuous swimming (Davenport, 1985). Because of the dual use of shelters, efforts are made to create shelters in salmon cages that accommodate both needs. The first shelters commonly used were variations on artificial seaweeds similar to those used for wrasses (see Skiftesvik et al., 2013). These shelters provided excellent options for hiding, but were not sturdy enough to provide suitable resting space (verbal feedback from salmon farmers). Other shelters, made from hard plastic, have provided good rigid resting space but have not provided much in terms of hiding places, as they are usually flat vertical surfaces, (see also Imsland et al., 2015).

Feed and shelter for lumpfish can become limited resources on a salmon farm. Feeding practices for lumpfish usually involve automated feeders dispersing feed in a limited area, though some farms use manual feeding. When feed is delivered manually, this is usually done once or twice per day with large quantities of feed delivered alongside the cage net and near the shelters. The fish keeper will look out for the fish and deliver more feed where the fish can be seen approaching the feed. Lumpfish are a docile and easily tamed species, so will usually approach fish keepers when they notice them. Though there is very little literature on how to best feed lumpfish, there is some indication that meal-time or “pulse” feeding promotes growth (Brown, Wiseman & Kean, 1997) and a more efficient feeding behaviour with less time spent foraging. On the contrary, fish may spend more time foraging and less time resting if feed is delivered continuously within a small area (Killen, Brown & Gamperl, 2007). Lumpfish will consume free floating feed pellets as well as pellets lying on the bottom of the tank (A Johannesen, pers. obs., 2015 from the production at the Nesvík Marine Centre). If shelters are located in such a way that a trade-off between an ideal resting/hiding place and feed availability emerges, there is a risk of competition for the best locations both in terms of shelter and feed (Pulliam & Caraco, 1984; Williams & Brown, 1991). Because of the potential competition related to the two resources, shelters and food, it is possible that shelter availability may negatively affect lumpfish growth or welfare if feed is a limited or difficult resource to gain access to. However, it is not currently clear whether continuous automated feeding provides better feed availability than manual “mealtime” feeding in this situation.

The aim of this study is to investigate the effect of shelters and feeding methods on lumpfish welfare, expressed by growth, body condition, and fin health. We use a crossed design with shelter availability: “Shelter” or “No shelter” crossed with two feeding methods: “Manual feeding three times per day” and “Automated feeding every 10 min during the day”. We expect that a shelter has a positive effect on growth and welfare and that manual feeding provides better growth due to prolonged resting times in between feeds. We also expect that “fin health” will be negatively affected by access to a shelter due to competition for optimal resting spaces, which causes a higher concentration of fish in a small area.

Methods

Setup

This experiment was carried out at Fiskaaling’s ‘Nesvík Marine Centre’ where lumpfish are experimentally bred for the purposes of establishing a domestic brood stock. Therefore, the fish used in this experiment were bred in captivity with a mixed ancestry of wild and captive bred brood stock and were accustomed to a captive environment with limited shelter access and a mixture of automated and manual feeding. For the experiment, 200 lumpfish (weight: 3.61 g ± 0.06; Mean ± SE) were taken from production tanks at the Centre and evenly distributed amongst 20 experimental tanks (10 fish per white fibre glass tank, 125 L, approximately 50 cm × 50 cm × 50 cm). Each fish was tagged using visible injectable elastomer (VIE) colour tags (Northwest Marine Technology Inc.) injected ventrally near the surface of the skin in two locations (Fig. S1). The duration of the experiment was 28 days, from 01-02-2017 to 01-03-2017 . At the beginning and end of the experiment, each fish was weighed in grams to two decimal points precision and total body length and height was measured to nearest mm. Additionally, all fish were assessed for fin damage on a scale from 1 to 4; 1: No damage (straight tail fin); 2: Moderate damage (small incisions on tail fin); 3: Severe damage, (large parts of fin missing, but without wounds); 4: Injury (missing tail fin and open wounds). Fish were anaesthetised using an aerated metacaine bath (0.2g/L) before handling to minimise stress. All tanks had flow through aerated sea water (7 °C), with a flow rate of two litres per minute or a full tank exchange every hour. Half lids provided partial cover while allowing for tank maintenance and overhead lights (121.00 lux ± 3.48 at surface) were set to a 12:12 light:dark schedule. Two experimental treatments were crossed to create four treatment groups with five tanks in each group. The treatments were; ‘feeding method’: (1) manual meal time feeding three times per day, (2) automated feeding for five seconds every ten minutes during the 12 light hours, and ‘shelter availability’: (1) shelter available, (2) no shelter available. Treatments were randomised amongst the tanks using the random number generator in Excel to account for any variation amongst tanks in light conditions and human disturbance during the day. Some fish were also photographed in order to document a difference in skin pigmentation that was noted upon measuring the fish at the end of the trial. One fish died of unknown causes during the trial period and has been excluded from analysis.

Feeding

The feed used in this experiment was 1.5 mm “Lumpfish Grower” feed pellets by Biomar, which is the standard feed used at Nesvík Marine Centre. These feed pellets are slow sinking with a protein content of 56% and lipid content of 15%. Since the lumpfish were accustomed to this feed, there was no adjustment period or change in their diet as a consequence of this experiment. Lumpfish can be expected to approximately double their weight every four weeks at normal growth. Daily feed amount was adjusted to be at least 5% of the expected final weight of the fish per day, which ought to be in excess of satiation. An excess of feed was delivered to ensure minimal competition and that insufficient daily feed quantity would not cause lowered growth in any group. Feed was delivered in the light period between 07:00 and 19:00 by automated feeders providing feed for 5 s every 10 min and manual meal times were at 08:30, 12:30, and 15:30. Automated feeders were regularly checked to ensure that they were feeding the necessary amount of feed and manual feed was divided into three meal times ensuring that no less than the minimum 5% of expected final biomass was provided in total (despite apparent satiation). Uneaten feed was removed once per day (in the morning before the first manual feed) as part of the daily tank cleaning routine. Uneaten pellets on the tank floor were used as an additional means of estimating whether sufficient feed was being delivered by the automated feeders.

Shelter

All tanks were white, cubical with flat bottoms, had a single green inlet pipe, and a central grey drainage pipe. The pipes offered some opportunity for substrate choice and potential colour matching in the unsheltered tanks, as lumpfish are often observed attaching to features in a tank contrasting with the background colour (Á Johannesen, pers. obs., 2013–2017). However, they were unavoidable in our setup and did not provide any opportunity for hiding. The shelters were constructed from square black PE drainage pipes cut in half lengthwise to create two right angle lengths of shelter (40 cm long, total area was 0.2 m2). These were hung horizontally (roof style, see Fig. S2) in twos, with one above the other and a gap between them of 20 cm. This construction allowed the fish to sit on top of, in between, and under shelters, allowing a large variation in degree of sheltering, from completely obscured from the top and sides of the tank to completely visible but slightly camouflaged by colour matching. Shelter usage was not recorded in a structured manner, but casual observation indicated that approximately 70% of the fish chose to sit underneath the shelters in such a way that they were not visible to people looking down into the tank.

Analysis

A mixed effects generalized linear model was constructed using ‘weight gain’ as the response variable, ‘shelter availability’ and ‘feeding method’ as the predictor variables, and ‘tank’ as a random effect. The analysis was carried out using the package “lme4” (Bates et al., 2015). A linear model was constructed to further investigate differences between the four treatment groups because the initial mixed model produced a significant interaction, making it difficult to interpret individual factor effects. Fish weigh, length and height was used to get an estimate of ‘body condition’ (weight (g)/length (cm) × height (cm)). Body condition at the end of the trial was used for constructing a mixed model with the same predictor variables as for ‘weight gain’, though a post-hoc test was not necessary as there was no interaction between the predictor variables. To investigate fin damage a different approach was necessary due to the more subjective measure of fin damage scores. Scores were saved as an ordered factor due to their non-linear nature. The package “ordinal” (Christensen, 2015) was used construct a cumulative link mixed model with the Laplace approximation for fin damage scores. Two models were constructed; one for change in fin damage score from the beginning to the end of the trial, and one for effect of treatment. All models used were mixed effects models with ‘tank’ as the random effect due to the dependent nature of individuals within the same tank, especially considering that within-tank competition might occur. All analyses were done in R Core Team (2017). For plotting, ggplot2 (Wickham, 2009) and ggthemes (Arnold, 2017) were used.

Ethical note

The work for this manuscript was reviewed by Fiskaaling’s “Animal Experimentation Ethics Committee” (approval number 002). The approval was based on the potential welfare benefit for lumpfish in aquaculture and the limited suffering expected to be caused by this study.

Results

In the 28 days that the trial lasted, mean weight increased by 4.87 g ± 0.14 (Fig. 1). Feeding method and shelter availability interacted (Chi-squared1,16 = 7.77, P = 0.005, Fig. 1). A post-hoc linear model revealed that the treatment group ‘shelter+automated’ had significantly lower increase in weight than all other groups (summary table comparison to ‘shelter+automated’; ‘shelter+manual’: t = 4.59, P < 0.001; ‘no shelter+automated’: t = 3.43, P < 0.001; ‘no shelter+manual’: t = 4.02, P < 0.001; model statistics: F3,195 = 8.52, P < 0.001; Fig. 1).

Figure 1 Change in weight in the 28 day trial period split into four treatment groups.

Boxes and whiskers describe quartiles for each group of 49 or 50 fish and dots are outliers.

Fin damage score improved significantly over the course of the experiment (Z =  −10.16, DF = 5, P < 0.001, Fig. 2) and the portion of fish with undamaged fins increased from 13% to 31% (Fig. 2). No effect of treatment was found on the fin scores at the end of the experiment (shelter: Z = 1.41, P = 0.158; feeding method: Z =  − 0.18, P = 0.854, Fig. 3).

Figure 2 Condition of lumpfish fins before and after the trial.

Bars represent the percentage of fish with each fin condition score. Scores represent: (1) no damage (straight tail fin); (2) moderate damage (small incisions on tail fin); (3) severe damage, (large parts of fin missing, but without wounds); (4) injury (missing tail fin and open wounds).

Figure 3 Fin condition scores after the experiment.

Bars represent the percentage of fish for each fin condition score split across treatments. Scores represent: (1) no damage (straight tail fin); (2) moderate damage (small incisions on tail fin); (3) severe damage, (large parts of fin missing, but without wounds); (4) injury (missing tail fin and open wounds).

Body condition at the end of the trial was significantly lower in tanks with a shelter compared to those without shelters (median body condition: 0.71 and 0.75 respectively; X1,52=4.20, P = 0.04) and significantly higher in tanks with manual feeding than in tanks with automated feeding (median body condition: 0.75 and 0.72 respectively; X1,52=5.33, P = 0.02). No interaction was found between the effects of treatments on body condition (X1,62=1.80, P = 0.18, Fig. 4).

Figure 4 Body condition of lumpfish at the end of the trial.

Boxes and whiskers represent quartiles for each treatment group and dots are outliers. Body condition was calculated from weight, length, and height of each fish: weight (g)/(length (cm) × height (cm)).

Discussion

In this experiment, we find that both the availability of shelter and the way feed is delivered to lumpfish have consequences for growth and body condition. In terms of body condition, the availability of shelter and automated feeding both reduce body condition such that the group that had lowest body condition was the fish that were provided with a shelter and an automated feeder. In absolute growth, there is a slightly different story, because while shelter availability does seem to lower growth, this is outweighed by manual feeding so that the fish with a shelter available did not have a lower increase in body weight overall compared to fish that did not have a shelter. In terms of fin health, there is no difference between the treatments, but all treatments seem to have been better than conditions in the stocking tanks as fin health improved over time from before the experiment to after. Ordinarily, aggression would be expected to be higher at lower stocking densities, and since stocking density was lower in the trial tanks than in the stocking tanks, a slight degradation in fin health was expected due to aggression and competition for shelter space. Improved fin health suggests that aggression was less prevalent in the trial tanks than in the stocking tanks. There may be other explanatory factors not investigated here. The fish were housed in a recirculation system before the trial and it is possible that the bacteria in that system were inhibiting fin healing, but there were no obvious signs of infection in any of the fish.

It is clear from this experiment that care needs to be taken to ensure that a situation of competition does not occur as a result of feeding method or deployment of shelters. Because of the behavioural need for a resting place and shelter (Imsland et al., 2015), it is important that suitable shelter is provided in sufficient amounts. However, because of a potential for competition for ideal spots in the shelter, care needs to be taken to ensure that feed is delivered in such a way that fish are able to compensate for any extra energy used for competition with increased feed intake (however, see Millidine, Armstrong & Metcalfe, 2006; Finstad et al., 2007).

Though enough feed was provided to all tanks to satiate all fish in this experiment, the amount delivered each 10 min from an automated feeder was so small, that not all fish had access to feed at the same time. Though not verified through quantitative observations, it is likely that the fish resting closest to where the feed was delivered were the fish that were able to feed first on any given day, and that the rest of the fish would feed only when the first fish reached satiation. It is also possible that a different temporal divergence occurred where dominant fish were able to feed at what was perceived as a less risky time causing lower ranked fish to limit their time spent foraging (Alanärä, Burns & Metcalfe, 2001). If fish closest to the feed delivery point could eat to satiation first, it is likely that there was competition for the spot in the shelter that was closest to where the feed was delivered. The competition for this spot may have caused increased use of energy and therefore lower growth than in the tanks with no shelter, where the bare tank surface area was large enough for all ten fish to sit near the feed delivery point (compare for example Andrew et al., 2004).

There is evidence in the literature that aggression increases when stocking density is low (Brown, Brown & Srivastava, 1992; Jørgensen, Christiansen & Jobling, 1993). Stocking density in this trial was very low, approximately 400 g per cubic metre compared to common stocking densities of 10–15 kg per cubic metre or more (Hosteland, 2017). It is possible that these results would not be replicated in a system with high stocking density due to decreased opportunity to establish social rank with resulting privileged access to feed. Therefore, the effects found here may be much diminished in a lumpfish production environment. However, in a typical salmon farm, the density of lumpfish is even lower than in this trial, approximately 10 g per cubic metre. This is a natural consequence of their small size and low numbers in salmon cages of 25–30 thousand cubic metres. Therefore, it is possible that situations will arise where lumpfish are able to establish a territory in specific locations in a salmon cage. Considering the need to adequately feed lumpfish in salmon cages, ensuring that all the fish have access to enough feed should be prioritised. As it can be difficult to track lumpfish in a salmon cage, one solution may be to disperse the lumpfish feed using an automated feeder similar to the commonly used salmon feeders and provide an additional daily meal of manually delivered feed near the shelters. Placing the shelters so that the feed can be delivered near them easily is worthy of consideration. However, there are other important considerations in terms of shelter placement that take priority over ease of feed delivery; these considerations being wave and sea current direction as well as shelter cleaning or maintenance.

In conclusion, while shelters can offer the necessary enrichment to meet the behavioural needs of lumpfish, careful planning is necessary when feeding methods are decided on. Growth is not necessarily a desirable lumpfish trait in a salmon cage, but body condition certainly is important for their welfare and survival. Similarly, for lumpfish producers, being able to provide optimal conditions for high growth allows producers to deliver lumpfish to salmon farms at a higher rate. Therefore, based on the results from this experiment, we cautiously recommend that feed is delivered in meal times in such quantities that all fish are able to feed simultaneously, especially in lumpfish rearing facilities. In salmon farms, manual feeding is best combined with automated pulse feeding using a feeder that disperses feed well. This ensures that feed is delivered when manual feeding is impractical, such as in bad weather or when days are much longer than working hours for staff.

Supplemental Information

Figure S1 Picture of lumpfish being tagged using VIE

The tags are a liquid elastomer that is injected into the surface of the skin. In a few hours, the tags harden to a rubbery texture. Fish were colour-coded using a combination of two out of five colours for each fish in each tank. This tagging method works well for lumpfish because they do not have scales that obscure tags. However, due to potential dark pigmentation, care has to be taken to locate tags where minimal pigmentation is expected. Photo credit: Nakita E. Joensen.

Click here for additional data file.

Figure S2 Tank with shelters

A tank with the two shelters hung as they were in all tanks with shelters. Upon careful inspection, it is possible to see some lumpfish sitting on the shelter. The remaining fish are most likely sitting underneath the shelter and so are obscured from view. Photo credit: Nakita E Joensen.

Click here for additional data file.

Video S1 Video recording of lumpfish cleaning behaviour

Video credit: Asa Johannesen.

Click here for additional data file.

Supplemental Information 1 Complete dataset for the results presented in this paper

Data are presented in ’long’ format, with measurements before and after in the same column and distinguished by a second column specifying whether the data point is before or after the trial. Weight is given in grams and length and height are given in centimetres.

Click here for additional data file.

Supplemental Information 2 Oxygen saturation data before and after the trial

Effluent oxygen saturation was measured at the start of the experiment and at the end. Measured values are provided in the sheet.

Click here for additional data file.

Additional Information and Declarations

Competing Interests

Author Contributions

Animal Ethics

Data Availability

The authors declare there are no competing interests.

Asa Johannesen conceived and designed the experiments, performed the experiments, analyzed the data, contributed reagents/materials/analysis tools, prepared figures and/or tables, authored or reviewed drafts of the paper, approved the final draft.

Nakita E. Joensen conceived and designed the experiments, performed the experiments, analyzed the data, authored or reviewed drafts of the paper.

Eyðfinn Magnussen conceived and designed the experiments, analyzed the data, authored or reviewed drafts of the paper.

The following information was supplied relating to ethical approvals (i.e., approving body and any reference numbers):

This work was approved by Fiskaaling’s “Animal Experimentation Ethics Committee” (approval number 002). The decision was based on the potential welfare benefits for lumpfish in aquaculture weighed up against methods that involved very little in terms of invasive procedures expected to cause suffering.

The following information was supplied regarding data availability:

The raw data are provided in the Supplemental Files.

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
