# Peer review of "Shelters can negatively affect growth and welfare in lumpfish if feed is delivered continuously"

_PeerJ, doi:10.7717/peerj.4837_

## Round 0.1 · original submission · Minor Revisions

We now have reviews back from two referees who are both enthusiastic about the value of the study, but each of which have a number of valuable suggestions for improvement that I encourage you to consider in revising your manuscript. In particular, it seems that the lack of information about water chemistry and details regarding the feed must be addressed in the revision as suggested by the referees. Beyond these specific issues, the advice of the referees seems reasonable and should be relatively simple to address, so I look forward to seeing your revised manuscript.

Reviewer 1 ·

Basic reporting

No comment

Experimental design

No comment

Validity of the findings

No comment

Additional comments

Review for “Shelters can negatively affect growth and welfare in lumpfish if feed is delivered continuously” (#25538)

The work outlined in this manuscript is an attempt at answering why high mortalities are occurring in lumpfish subjected to salmon cages. Although it is not directly applicable to a net pen scenario, given the experimental design are with tanks containing lumpfish only for a 28 day experiment, the results observed have merit. With some minor revisions and clarifications, I would recommend for publication.

Line #20. Add scientific name for Atlantic salmon
Line #45. Delete “much of this is”
Line #59. Delete “not mentioned much in literature, but”
Line 60 – 61. Please clarify. Is it not enough because of structures are not sturdy or because not enough space for the number of cleaners?
Line #100. What was the flow rate?
Line #110. Feeding. Only pellets are mentioned but no description of the feed. What was the brand used, pellet size, floating or sinking, percent protein & fat? Was it the typical feed used for lumpfish in captivity?
Line 117. Uneaten food was observed on the tank bottom. Was this uniform throughout all treatments? Clarify. Why would there be food on the tank bottom if fed properly?
Lines 174 – 175. The fact that fish with shelter had dark pigmentation compared to those without is interesting but unless you explain what this could mean in the discussion, it doesn’t have any significance. Delete.
Lines 186 – 187. Explain why fin health improved
Line 201. “or perhaps some other temporal divergence occurred.” Useless comment unless backed up. Explain
Line 209/212. The biomass in the tanks were said to be low density – 400 g/m3. Cages are considered low density at 10g/m3. How can both be low density when the experimental tanks are 40x that of cages? Reword.
Figure 2. Delete. Improved fin condition is mentioned in the text and the reader can also see it clearly in Figure 3.
Figure 4. Correct “Bocy” to “Body”
Figure 5. Delete unless explained significantly in text.

General comments:
There was no mention of water chemistries measured during this experiment. Even though flow through systems, with the uneaten feed left overnight on tank bottom, you cannot rule out water quality effects, especially if it was more prominent in one treatment group vs. another. Water chemistry parameters such as dissolved oxygen, total ammonia nitrogen, and nitrites should be provided.

The introduction is an excellent review of past and current methodologies for the removal of sea lice. However, to the general reader, a further introduction of how exactly the lumpfish clean the parasites off salmon would be beneficial. For instance, this experiment does not take into account the energy spent cleaning the salmon. Those that clean more do not need to feed pellets as much or do they need to be fed more often to compensate for energy cleaning salmon? It is assumed the lumpfish eat the parasites for nutrition? If low densities of lumpfish are in the cages, is there not enough of a parasite load to satiate the lumpfish’s diet? Alternately, in high parasite load environments, are lumpfish satiated so that methods of feeding inconsequential? This needs to be explained better in the introduction for the reader.

What is meant by manually feeding both in the cages and during this experiment? Is it typically done by broadcasting or placed in the same area at all times? Wouldn’t broadcast feeding prevent shelter effects?

·

Basic reporting

This article is clear and unambiguous. I found the article easy to read and follow.

In the Introduction I suggest the authors include a reference or two additional alternative treatments the industry has explored or continues to explore in treatment possibilities for sea lice. The industry has used garlic, and even modern technical approaches such as “lice lasers”. This would give a better sense to the reader on how devastating sea lice are in the aquaculture industry and the approach is to “try and investigate everything that kills lice”. Lumpfish of course, as presented by the authors, representing one of these possible approaches.

The article is a professional article and structure, figures/tables, and raw data were available.

The article is also self contained with relevant hypothesis.

Experimental design

This research is within scope of the journal. The research question is well defined, relevant, and meaningful.

Lumpfish are a relatively unexplored species for natural or mechanical removal of sea lice. There is also little literature available on the welfare or captive care of lumpfish. Captive care of lumpfish needs to be quantified better. Relying on wild brood stock is not sustainable in the long term and any information that contributes to the understanding of captive care is useful.

The authors do mention why fish farmers shun more “natural” refugia like artificial kelp. There are many exciting possibilities for future research exploring this and the impacts on lumpfish health (feeding) with host fish present.

The majority of the methods described in the article present with sufficient detail to replicate the process. However, please clarify your thoughts related to the random aspect of treatments, particularly human disturbance of tanks. For example, why did you not open the automatic feed tank lids to simulate human disturbance e.g. at 8:30; 12:30: 15:30 when you were also feeding the manual tanks? Are the fish used already habituated to human disturbance and raised completely in captivity? You mention they are taken from “Nesvík Marine Centre”. Do you find these fish habituate to human feeding and thus the fish you used for this experiment moved from human feeding to automatic feeding might have had a period of adjustment – thus consuming less food?

Please describe the feed type in detail. Is this the same feed these fish were raised on previously? From what I can gather in the literature lumpfish have a “blind spot” for feeding and will only eat suspended feed. Did you use a partially buoyant feed? A note (in methods) or a comment on the feeding behavior in the discussion related to this feeding blind spot would be useful. For example, perhaps lumpfish in shelters could not see the feed suspended in the water column as well as control lumpfish or the feed sank faster when touching the shelters. Did any feed end up on the shelters themselves (only the bottom is mentioned)?

Validity of the findings

The findings are valid as presented and not repetition of previous studies.

The conclusions are generally well stated and not over – reaching, if anything, the focus should be on improving captive care of lumpfish through examination of the feeding dynamics and body condition.

However, a wider connection to the salmon farming industry is merited. The authors state “Therefore, based on the results from this experiment, we cautiously recommend that feed is delivered in meal times in such quantities that all fish are able to feed simultaneously.” I appreciate being cautious but if I were a fish farmer/reader trying to understand this I would remain uncertain. If I stock lumpfish in my pen, I don’t actually know where they are, and I can’t track them. I’d suggest the authors, if they wish, comment more directly on what “advice” they would give to salmon farmers on shelter placement, and feeding manually versus automatically. However in doing so, they state that the evidence in their research applies primarily to captive care or rearing of lumpfish.

Additional comments

I found the overall paper well written and not over reaching in their conclusions. If anything, they could speculate on their advice to the average fish farmer working in a tank based rearing system (cultivating lumpfish) and compare this directly to a fish farmer raising salmon in netpens.

---

## Round 0.2 · accepted · Accept

Both referees agreed that the original submission was well-written, clear and of considerable value to the field. Their recommendations for additional details have all been followed in the revised submission, and in particular the specifics of the water flow, quality and feed have been added to the text or supplementary materials as requested. I find the revisions to be comprehensive and I can see the authors point in the few minor cases where they disagree with the suggestions of the reviewers. I believe that you have adequately addressed the relatively minor concerns of the referees from the first round of submission, and see no reason to return it for additional review. Therefore, I am happy to recommend the paper for acceptance and move it forward into production.

#